# Hydrophobic Recovery of Plasma-Hydrophilized Polyethylene Terephthalate Polymers

**DOI:** 10.3390/polym14122496

**Published:** 2022-06-19

**Authors:** Gregor Primc, Miran Mozetič

**Affiliations:** Department of Surface Engineering, Jožef Stefan Institute, Jamova Cesta 39, 1000 Ljubljana, Slovenia; gregor.primc@ijs.si

**Keywords:** PET, oxygen plasma, VUV photons, hydrophilicity, wettability, hydrophobic recovery

## Abstract

Oxygen plasma is widely used for enhancing the wettability of numerous polymers, including polyethylene terephthalate (PET). The treatment with plasma containing oxygen will cause surface functionalization with polar functional groups, which will, in turn, improve the wettability. However, the exact mechanisms leading to the hydrophilic or even super-hydrophilic surface finish are still insufficiently explored. The wettability obtained by plasma treatment is not permanent, since the hydrophobic recovery is usually reported. The mechanisms of hydrophobic recovery are reviewed and explained. Methods for suppressing this effect are disclosed and explained. The recommended treatment which assures stable hydrophilicity of PET samples is the treatment with energetic ions and/or vacuum ultraviolet radiation (VUV). The influence of various plasma species on the formation of the highly hydrophilic surface finish and stability of adequate wettability of PET materials is discussed.

## 1. Introduction

Polymers, polymer blends, and polymer composites are widely used materials. The type of polymer useful for a particular application is chosen according to its chemical and mechanical properties. The surface properties of the selected material, however, are rarely adequate. Among the most desirable surface property is the wettability, i.e., the ability of a polymer to be wetted by liquids. The nonpolar liquids usually wet most polymers, but the wettability for polar liquids is often inadequate, since the deposited liquid tends to form droplets upon deposition to the polymer surface rather than a uniform thin film. The reason for the spontaneous organization of a deposited liquid into numerous droplets is the discrepancy in the surface energy of the liquid and the substrate. The droplets will be formed if the substrate’s surface energy is much lower than the surface energy of the liquid. The surface energy has two components, i.e., the dispersive and the polar component. The dispersive component arises from the simple fact that the molecules on the polymer surface are bound to the bulk only on one side. The polar component obviously depends on the polarity of the surface molecules or their fragments. Highly polar functional groups are those containing oxygen. Namely, the oxygen atoms will attract a negative charge, while the rest of the molecule will be positively charged against the oxygen-containing molecular fragment. Taking into account this fact, the oxygen-containing polymers will exhibit better wettability than the oxygen-free polymers.

Wettability is often determined by one of the most straightforward experimental techniques—deposition of a small water droplet. The wettability will be good if the water droplet spreads on the surface and exhibits a small contact angle. Figure 1 represents selected polymers’ reported water contact angles (WCA). The WCA is plotted versus the ratio between the amount of oxygen and carbon atoms in the polymer (O/C). While there is an apparent trend (the WCA decreasing with increasing oxygen content), the results are scattered significantly. Furthermore, none of the known polymers exhibit very good wettability (WCA below 20°). The O/C ratio in the as-synthesized polymer is not a decisive parameter governing the polymer wettability. The paradox is explained by the basic principle of thermodynamics: the oxygen-containing groups do not stretch from the polymer surface but are rather oriented towards the bulk. Such an orientation will minimize the free energy to the most stable state of a polymer.

The surface properties of polymers should be modified in many applications. In many cases, short-term surface modification is adequate or at least tolerated, but, in some other applications, the long-term stability of the modified surface is desired. Taking into account the upper considerations, the preferred method for increasing polymer wettability is functionalization with oxygen-containing functional groups that stretch from the surface. Among methods for such surface decoration with oxygen groups, treatment with non-equilibrium oxygen plasma is the most straightforward approach. A polymer sample is briefly exposed to oxygen plasma, and the reactive plasma species (positively charged ions, radicals in the ground, and metastable states) interact chemically on the polymer surface to break the C-H, as well as C-C, bonds and occupy the dangling bonds. This is a very simplified illustration of plasma-induced functionalization of polymers with oxygen-containing functional groups. Details will be described below in this manuscript.

As already mentioned, the high concentration of polar surface functional groups is against the rules of thermodynamics, so the plasma effect is not permanent. The slow decay of the polymer wettability is often called “hydrophobic recovery” [2]. The hydrophobic recovery of plasma-treated polymers was tackled by numerous authors, who proposed the following mechanisms:Segmental mobility of various sequences [3,4,5,6];Gasification (desorption) of the low-molecular-weight fragments [7,8];Reorientation and migration of polymer chains [4,5,6,7,8,9,10,11,12,13,14,15];Short-range segmental motion, hiding polar functional groups from the surface [11,14,16,17,18];Disappearance of some surface groups [12,19];Surface contamination with adsorbed organic molecules [14,17,18,20];Neutralization of trapped charges [18].

The mechanism may be specific to some polymer materials and plasma treatments. In any case, various mechanisms have been proposed, and some elaborated in detail. Little work, however, has been conducted on the theoretical calculations on the atomic or molecular level.

The new idea of the work is to examine the peculiarities of the plasma systems used by various authors to hydrophilize PET polymers. Most authors have not measured plasma species’ fluxes and/or fluences and their energies but reported on the discharge parameters. A skilled plasma scientist will be able to estimate the fluxes from the reported peculiarities of the systems and, thus, identify the predominant species which cause the surface modifications. The penetration depth of different species differs, so it is possible to deduce the necessary treatment conditions that lead to the modification of a thicker surface film, which was found to be beneficial for the suppression of the hydrophobic recovery. This article focuses on the hydrophilization and hydrophobic recovery of polyethylene terephthalate and summarizes the results reported by various authors. We correlate the material wettability with the processing parameters based on available data. Such correlations have not been reported in the scientific literature, thus presenting another novelty of this article.

## 2. Hydrophilic and Super-Hydrophilic Polymers

Hydrophilic polymers are often referred to as polymers that either dissolve in water or are swollen by the water. The hydrophilic polymers are naturally produced and include many polysaccharides and proteins (for example, alginate and albumin). Hydrophilic polymers can also be manufactured, such as polyvinyl alcohol. The hydrophilic polymers are useful for some applications because of the solubility in water, but this property is a drawback in many other applications. The insoluble polymers are obviously hydrophobic if solubility in water is the merit.

The hydrophilic character of the polymer, as defined above, should not be misinterpreted when considering the surface property of a product made from a polymer or containing polymers. The surface hydrophilicity is a standard expression adopted for describing the surface wettability, i.e., the ability of a material to enable the spreading of a polar liquid (such as water) on a large surface. The term “surface wettability” does not refer to solubility in water, but the expression “hydrophilic polymer” is often used in scientific literature to describe the surface finish. The praxis is also adopted in this article.

The surface hydrophilicity (or, simply, “hydrophilicity”) is probed by careful deposition of a small droplet of a polar liquid on the surface and measuring the contact angle with the solid material. The droplet should be small enough to assure the negligible influence of the deviation from a spherical geometry due to the gravitation when the ratio of the surface energy of the liquid and solid approaches infinity. In practice, a water droplet of a few microliter volumes would satisfy this requirement.

The shape of the water droplet on hydrophobic and hydrophilic polymer surfaces is illustrated in Figure 2. Figure 2a illustrates a large water droplet on a highly hydrophobic surface. The flattening of the shape is due to the gravitational force, which tends to minimize the gravitational potential energy. When the gravitation force is marginal as compared to the surface tension, the flattening vanishes, and a small droplet of a polar liquid on the superhydrophobic solid surface assumes spherical geometry, as illustrated in Figure 2b. The minimization of the droplet volume will cause an infinitely low contact area between the droplet and the surface. Figure 2c shows a water droplet on a hydrophilic surface; the contact area is large, but the water does not spread on a very large surface because of the final ratio between the surface energies of the liquid and the solid. The solid material is superhydrophilic when the contact angle of the water droplet is very low, as illustrated in Figure 2d. The superhydrophilicity is often referred to as the water contact angles below a few degrees.

The difference between highly hydrophilic and superhydrophilic surfaces is illustrated in Figure 3. The latter never occurs on smooth surfaces, regardless of the surface composition (Figure 3a). The superhydrophilic surface is observed only for materials of rich morphology on the sub-micrometer scale. The reason is the dragging of liquid by capillary forces. When a liquid droplet is deposited on the surface of a superhydrophilic material, it will fill the gaps, holes, or similar morphological features (Figure 3b), so the contact angle of a water droplet will be difficult to measure. The necessary conditions for a material superhydrophilicity are, therefore, twofold: 1. the presence of highly polar surface functional groups, and 2. rich morphology on the sub-micrometer scale. The critical conditions probably depend on the peculiarities of the material and are yet to be reported in the scientific literature.

## 3. Surface Kinetics upon Treatment of Polyethylene Terephthalate with Oxygen Plasma

Polyethylene terephthalate (PET) is the most common thermoplastic polyester, with an annual production of over 30 million tons [21]. Its application is broad, from textiles to engineering plastics. Recent applications include flexible light-emitting diodes because of their good mechanical deforming ability, smooth surface, good thermal durability, and reasonably high oxygen and moisture blocking capability [22]. However, the surface properties of the as-synthesized PET foils prevent the application in doping-free white organic light-emitting diodes with high color rendering [23,24]. A monomer unit consists of an aromatic ring with two ester groups. The chemical formula is (C_10_H_8_O_4_)_n_. Despite the relatively large percentage of oxygen, the PET is a moderately hydrophobic polymer with a static contact angle between 70 and 80° [25]. The WCA is often decreased by a brief treatment with a nonequilibrium gaseous plasma sustained in gas that contains oxygen. It could be almost pure oxygen [26,27,28,29,30], oxygen mixed with a noble gas [31,32], or air [30,33,34]. Some authors also probed treatment with a plasma sustained in “pure” noble gases [27,35] or vacuum ultraviolet (VUV) irradiation [36,37]. Modification of the surface properties of PET upon treatment with gaseous plasma is a result of interaction between the polymer and neutral reactive species, positively charged ions, and radiation. The effect of plasma electrons and negatively charged ions (such as O_2_^−^ and O^−^) can be neglected in most cases, since the polymer samples are usually kept at floating potential upon plasma treatment. The positively charged ions are accelerated towards the surface and all negatively charged particles are decelerated and retarded by the electric field within the sheath. The penetration depth of neutral radicals, positively charged ions, and VUV radiation from nonequilibrium plasma in the polymer samples is shown in Figure 4.

The neutral radicals are thermal in plasma. They assume a Boltzmann distribution over the kinetic energy with the average kinetic energy corresponding to the neutral gas kinetic temperature (gas temperature). At the gas temperature of 300 K, the average kinetic energy of neutral particles is about 0.04 eV. Even at the temperature of 1000 K, the average kinetic energy is about 0.1 eV, so marginal as compared to the binding energy of atoms in PET. A lack of the kinetic energy of neutral plasma species limits the interaction with the solid material to pure potential interactions, like chemical binding to the uppermost carbon atoms and heterogeneous surface recombination.

Positively charged ions are accelerated in the potential fall between plasma and the polymer surface. Depending on the electron temperature in nonequilibrium gaseous plasma, the positively charged ions gain certain kinetic energy in the sheath and accelerate perpendicular to the polymer surface. Depending on the gas pressure, the sheath is either collisionless, where practically all positive ions are monochromatic, or there are collisions that distort the perpendicular motion of ions and also suppress the average kinetic energy. If the polymer is kept at the floating potential, the kinetic energy in the collisionless approximation is roughly 10 eV. The positive ions of such kinetic energy are able to penetrate only a few monolayers within the polymer. On the other hand, in cases when a polymer sample is placed on the electrode powered with a radiofrequency generator, the DC self-biasing causes significant acceleration of positive ions. The DC voltage is about half of the RF peak-to-peak voltage, which is still several 100 V. In the collisionless sheath approximation, the positive ions bombard the polymer surface with the kinetic energy of the order of 100 eV, which is much larger than the binding energy between atoms in the polymer.

Gaseous plasma glows, so it is a source of radiation. The radiation in the visible part of the spectrum is usually negligible compared to the radiation intensity in the VUV part of the spectrum. VUV photons are evenly distributed over the solid angle and can penetrate the polymer sample. The penetration depth for VUV photons arising from oxygen plasma in PET polymer is yet to be reported in the scientific literature. Still, according to measurements performed with the interaction of VUV and other polymers, one can deduce the penetration of photons of the primary O-atom line at 130 nm to about 50 nm [38]. The pioneering work on the influence of VUV radiation on the PET surface composition was published by Hollander et al. [39]. They used a few VUV sources and found either an increase or decrease in the O/C ratio with increasing treatment time, depending on the range of wavelengths specific to the sources. In all cases, the irradiation caused the formation of low molecular weight compounds, many desorbed from the surface under high-vacuum conditions [39]. Prolonged treatment (up to 1 h) caused significant cross-linkage and discoloration due to the formation of double bonds. The addition of oxygen into the experimental chamber caused rapid etching. More recently, Zhang et al. [40] reported on the evolution of PET surface and sub-surface properties upon treatment with VUV radiation in the range of photon fluences between 10^20^ and 10^22^ m^−2^. They found preferential cleaving of the oxygen-containing functional groups at relatively low doses and significant photolysis at prolonged treatment.

Figure 4 illustrates the penetration depth for various reactants. Let us focus on neutral reactive plasma species suitable for PET hydrophilization, i.e., atomic oxygen and OH radicals. Both exhibit a considerable oxidation potential, and any interaction between these radicals and a polymer surface is exothermic. The theory explaining the interaction of PET with neutral radicals is yet to be elaborated. According to the recent theory for oxidation of polystyrene with O-atoms [41], there are over 10 binding sites on the pristine polymer surface, and the most energetically feasible reaction is the substitution of a hydrogen atom in the C-H bond with the hydroxyl radical to form C-OH surface functional group. By increasing the O-atom fluence, other oxygen-containing functional groups also appear on the polymer surface. Kinetics of the appearance of various oxygen-containing functional groups on PET surface upon exposure to O-atoms is yet to be elaborated, but the concentration of oxygen in the surface film as probed by XPS versus the O-atom fluence was reported by Vesel et al. [25]. The authors probed a broad range of fluencies between 10^20^ and 10^25^ m^−2^ and found a roughly logarithmic increase in the oxygen concentration. The observation can be explained either by a slow increase in the concentration of surface functional groups, diffusion of the oxygen inside the polymer within the layer of thickness of the XPS probing depth (several nm) [42] or by increasing surface roughness. The surface roughness versus the fluence of O-atoms was reported by Junkar et al. [28]. They used very smooth PET foils with the *R*a parameter (as determined by AFM on the 3 µm × 3 µm surface) below 1 nm. The *R*a increased with increasing O-atom fluence and stabilized at about 10^25^ m^−2^. An identical behavior was reported for the wettability: the WCA at first decreased with increasing O-atom fluence but stabilized at about 15° after the sample was exposed to the fluence of 10^25^ m^−2^. The authors concluded that the surface modification of PET with O-atoms was accomplished at the O-atom fluence of 10^25^ m^−2^. Further treatment only causes etching, but the wettability, surface composition, and morphology remained unaffected. The etching of PET by O-atoms was elaborated by Doliska et al. [29], who reported the etching rate of about 3 nm/10^25^ m^−2^. Such a low etching rate was measured by a quartz crystal microbalance. The interaction of O-atoms with PET is illustrated in Figure 5. The functional groups are concentrated on the surface. A slight diffusion of oxygen into the subsurface layers was reported by Le et al. [42] and Zhang et al. [40], but the polar functional groups were concentrated on the very surface of the PET samples.

Plasma consists of free electrons and positively charged ions. Negatively charged ions are also formed in plasma of electronegative gas, such as oxygen. The electrons are much faster in plasma than other particles, so they quickly reach any object’s surface, including a PET sample. The polymer surface is, therefore, negatively charged. Due to the repulsing electrostatic force, the negative charge is limited to the surface (does not enter the bulk). The charge accumulated on the surface will retard all plasma electrons except those from the high-energy tail of the electron energy distribution function. The equilibrium concentration of the negative surface charge is obtained when the fluxes of positively and negatively charged particles from the plasma onto the surface are equal. As long as the kinetic energy of ions impinging the PET surface is small (the polymer is at floating potential), the effect of ions on the surface finish can be neglected, and the surface finish is similar to that illustrated in Figure 5. When the polymer is negatively biased, however, the ions will cause extremely rich surface morphology, as they possess large kinetic energy. PET is a nonconductive polymer, so the negative bias is achieved by placing a polymer foil onto a powered electrode of a capacitively coupled radio-frequency discharge. A schematic of the interaction of positively charged oxygen ions with the PET surface is illustrated in Figure 6. No numerical values for the ion fluences have been reported in the scientific literature, so the y-axis of Figure 6 has arbitrary units.

Exposure to energetic ions causes the formation of etching inhibitors, which may be islets of polymer damaged by impinging ions [43] or deposited foreign material. The role of inhibitors was elaborated by Gogolides’ group, unfortunately not for PET [44]. The etching inhibitors self-organize in islets with the lateral dimension of the order of 10 nm [19]. The PET surface exposed to oxygen plasma with DC self-biasing at about −400 V causes the formation of nanopillars whose diameter is not affected much by the duration of plasma treatment, but the height increases almost linearly with the increasing treatment time for the first 10 min, as shown by Oh et al. [19]. Such a PET surface becomes super-hydrophilic, i.e., exhibits immeasurably low WCA. The super-hydrophilic surface finish of PET was also observed for samples kept at the floating potential upon treatment with oxygen plasma sustained by electrodeless radiofrequency discharge [45], so it seems that the appropriate nanostructuring occurs even at the low kinetic energy of oxygen ions in a limited range of discharge parameters. Still, the recipe for rich surface roughness and super-hydrophilic surface finish is the treatment of PET materials with oxygen ions of the kinetic energy of the order of 100 eV.

## 4. Hydrophobic Recovery of Polyethylene Terephthalate and Methods for Suppressing This Effect

The hydrophobic recovery of plasma-activated PET was studied systematically by several groups. Lopez-Santos et al. [31] activated PET samples using a beam of oxygen atoms from a commercial source—a magnetized microwave discharge. The O-atom kinetic energy was supposed to be below 1 eV. The flux of O-atoms on the PET surface was as low as 5 × 10^17^ m^−2^ s^−1^. The treatment with O-atoms for half an hour resulted in a WCA of 43°. The aging was performed under ambient conditions, and the WCA after a month was about 68°. The same group also used an atmospheric pressure DBD discharge sustained in argon with oxygen admixture and reported a WCA of about 50° after treating the PET foil for half a minute. Complete hydrophobic recovery was reported in the case of PET activation by plasma sustained by the DBD discharge. The authors also probed a remote microwave plasma sustained in Ar/O_2_ mixtures at the discharge power of 60 W and pressure of 45 Pa. The WCA dropped to 24° after a 1-min treatment. One month of aging resulted in an increase in WCA to 56°. XPS characterization of samples revealed a significant decrease in the O/C ratio for all cases.

Junkar et al. [45] used an inductively coupled radiofrequency discharge at the power of 200 W for sustaining oxygen plasma at the pressure of 75 Pa. They reported quick functionalization with polar functional groups and gradual nanostructuring, which led to the super-hydrophilic surface finish at the treatment time of about a minute. The samples treated for 90 s were tested for hydrophobic recovery. The samples were kept at ambient conditions and room temperature, and the aging was monitored for two weeks. The WCA increased from an immeasurably low value to several degrees within a few hours and stabilized at about 20° after a week.

Jucius et al. [46] sustained RF oxygen plasma at the power density (power per unit surface of the electrode) of 300 W/m^2^ at the pressure of 133 Pa. The total discharge power was about 500 W, and the plasma treatment time was varied up to a minute. The authors observed a gradual decrease in the WCA with increasing treatment time. The WCA was 36° after treating samples for 10 s, 27° after 20 s, and 18° after a minute of plasma treatment. The hydrophobic recovery was examined at ambient conditions after 4, 10, 30, and 60 days. The hydrophobic recovery occurred within the first three days, and only a marginal increase in the WCA was observed thereafter. The WCA after 60 days was 54°. This group also probed a subsequent treatment of PET with oxygen and CF_4_ plasma and observed an interesting aging effect: the WCA of the as-treated samples was about 92°, but it dropped to about 20° after 3 days of aging. This value remained stable even after 60 days of aging under ambient conditions. The rather unexpected results were explained by the formation of low molecular weight fragments after the treatment with CF_4_ plasma. The fragments were supposedly desorbed upon aging. The authors, thus, provided a recipe for stable hydrophilicity of PET samples by using successive O_2_ and CF_4_ plasma treatments.

Oh et al. [19] studied the hydrophobic recovery of PET samples treated in oxygen plasma at the pressure of 2.6 Pa. The samples were biased to −400 V, but details about the discharge configuration were not reported. The plasma-treated samples acquired a super-hydrophilic surface finish. The aging was performed at different temperatures of RT, 40, 80, and 130 °C. The aging time was up to 24 h. They probed both amorphous and semicrystalline PET. The hydrophobic recovery at room temperature was marginal for both materials. At 40°, the recovery was faster for amorphous PET. Aging at 80 °C caused a rapid hydrophobic recovery for both types of PET and a superhydrophobic surface finish for some samples. The superhydrophobic character of samples aged at 130 °C persisted for semicrystalline samples, but the amorphous samples gradually lost the very large WCA at such a high temperature.

Perez-Roldan et al. [27] used an asymmetric capacitively coupled RF discharge for sustaining oxygen plasma at the discharge power of 75 W. The oxygen pressure was only 0.033 Pa. Samples were placed onto the grounded housing, so they were kept at the floating potential during the plasma treatment. The WCA dropped to about 20° after a minute of plasma treatment and slowly decreased with additional treatment time until an immeasurable low value was observed after half an hour. The roughness of the samples was also found to increase gradually with increasing treatment time and stabilized at about *R*a = 4 nm (as determined by AFM) after half an hour. The surface finish was, therefore, similar to that illustrated in Figure 5. The hydrophobic recovery was studied systematically for samples treated for different plasma durations and up to one month of aging. Interestingly enough, the samples treated for longer times exhibited much more pronounced aging than samples treated for a few minutes only. The sample treated for a minute assumed a final WCA of 60° after a month of aging, while those treated for half an hour exhibited the WCA of almost 90°, so about 10° more than the pristine PET. The results were explained by the loss of surface functional groups but the preservation of the morphology.

Chen et al. [47] used a capacitively coupled RF discharge to sustain oxygen plasma at the pressure of 60 Pa. The authors mounted PET samples perpendicular to the electrode to avoid bombarding the polymer foil with energetic positively charged ions. They probed different discharge powers in the range between 20 and 100 W and treatment times up to 10 min. They found the power of 80 W and treatment time of 2 min most useful and studied the hydrophobic recovery under ambient conditions in air and water. The WCA of the as-treated samples was about 36°, and the aging was negligible over the time of up to 90 days. Namely, the WCA increased only by about 10°. The aging in water was more rapid, and the WCA stabilized at almost 60° already after a few days of storage.

Inagaki et al. [26] treated PET samples in a glass discharge tube where the plasma was capacitively coupled with an RF generator. Oxygen pressure was 13.3 Pa, the discharge powers 25, 50, and 100 W, and the treatment time was up to 3 min. Aging experiments were performed upon storage in a temperature-controlled desiccator at 25 °C for up to 30 days. Unlike other authors, Inagaki also provided some correlations between the WCA, discharge power, etching rate, weight loss, etc. The WCA decreased gradually with increasing treatment time and assumed the values of 40, 26, and 22° after treating for 180 s at the discharge powers of 25, 50, and 100 W, respectively. Despite the relatively rich morphology and surface finish, as illustrated in Figure 5, the super-hydrophilic surface was not achieved since the lowest reported WCA was just above 20°. Gradual hydrophobic recovery was reported, and the WCA assumed about 50° after about two weeks of aging and remained constant thereafter.

Atmospheric pressure plasmas are also popular for surface activation of PET samples. Borcia et al. [48] treated samples for 0.1 s and reached the WCA of 52°. The aging was studied for two weeks under ambient conditions, and the WCA increased to 65°. They also treated the samples for 5 s and observed the WCA of 38° just after the treatment and 58° after two weeks.

Homola et al. [49] used a coplanar version of the DBD with the discharge power of 400 W and the surface energy density and volume energy density of approximately 2 W cm^−2^ and 80 W cm^−3^, respectively. They observed a rapid decrease in the WCA during 1 s of plasma treatment, and the WCA was, after that, relatively constant at about 37°. The hydrophobic recovery was accomplished within a few hours, and the WCA stabilized afterward at the values of about 52° and 59° for samples treated for 1 and 10 s, respectively.

Gotoh et al. [50] sustained a plasma jet in ambient air using a pulsed generator operating at 285 V and a current as large as 6 A. The treatment time was varied between 0.025 and 0.125 s to avoid melting the PET samples upon treatment with the powerful plasma. The WCA of the as-treated samples was slightly dependent on the treatment time and distance between the nozzle and sample, but always between 20 and 30°. Aging was studied under ambient conditions. Hydrophobic recovery was observed only for the advancing water contact angle, and the receding WCA remained constant. The static WCA kept increasing up to the longest aging time of 2 weeks, so it is likely that the hydrophobic recovery was not accomplished during that period.

In another paper, Gotoh et al. [36] used the same plasma to modify PET surfaces as in [50], except that they also reported the application of VUV radiation arising from a xenon excimer lamp operating at the wavelength of 172 nm. The irradiation time was 60 s, and the power density of the VUV irradiation on the sample surface was 16 mW/cm^2^. The observed advancing WCA of the as-irradiated sample was rather large at 46°, but remained unchanged even after a week of storage at ambient conditions. They compared the hydrophobic recovery of VUV and plasma-treated samples to prove that the VUV irradiated PET did not exhibit any hydrophobic recovery.

The explanation for the results reported by Gotoh et al. [36] was provided by Hozumi et al. [16]. Hozumi and colleagues exposed a polystyrene foil to the same type of xenon excimer lamp as Gotoh et al. [36] for various periods and studied the evolution of wettability. They did not use any plasma treatment; instead, they placed the polymer samples into a vacuum chamber filled with pure oxygen at various pressures. Independently of the oxygen pressure, the polymer samples assumed a super-hydrophilic surface finish, but the required treatment time depended on the oxygen pressure. At atmospheric pressure, super hydrophilicity was observed at the VUV treatment time of less than 50 s, while, at 10 Pa, it took about 15 min. The authors explained the rapid hydrophilization with a combined effect of bond scission by VUV and enhanced functionalization by atomic oxygen formed upon photodissociation. The surface functionalization was studied by angular-resolved XPS. Hozumi et al. clearly showed that the functionalization was surface-limited at the low oxygen pressure, in contrast to high pressure, where the O/C ratio was practically independent of the photoelectron takeoff angle. Evidently, the irradiation at a large concentration of oxygen in the experimental chamber enabled uniform functionalization of the surface layer of thickness at least equal to the escape depth of photoelectrons, i.e., several nm. The aging under ambient conditions was studied for a month. While the samples irradiated at low oxygen pressures exhibited rapid hydrophobic recovery, which is typical for plasma-treated polymers, the samples irradiated in oxygen at atmospheric pressure did not show any aging. Therefore, Hozumi et al. [16] provided a recipe for long-term stable hydrophilization of polymers: irradiation with VUV in oxygen at atmospheric pressure. As mentioned earlier, Gotoh et al. [36] confirmed the applicability of this technique also for PET.

The results of hydrophobic recovery of oxygen plasma-treated PET, as reported by the above-cited authors, are summarized in Table 1. Correlations between the surface finish and the processing parameters are plotted in Figure 7, Figure 8 and Figure 9. Figure 7a is a plot of the temporal evolution of the WCA upon storage, as reported by all authors mentioned in Table 1. The x-axis scale is linear. Rapid hydrophobic recovery is observed within the first hour of polymer aging, so the evolution of the WCA is better viewed on the logarithmic scale (Figure 7b).

Figure 8 represents the reported WCA of as-treated PET samples versus the pressure in the plasma chamber. The results are scattered significantly, which indicates that the pressure is not a decisive parameter. Still, the trend is noticeable: the WCA of the as-treated samples increases with increasing pressure. In fact, the hydrophilicity of samples treated at atmospheric pressure is relatively poor—the average value of the WCA is just below 40°. Reasons for such inadequate surface finish of samples treated at atmospheric pressure are yet to be elaborated. One possible explanation would be the low kinetic energy of positively charged ions impinging on the polymer surface. The sheath between plasma and substrate is rarely collisionless at atmospheric pressure due to the short mean free path. Despite the acceleration in the electric field, ions collide with neutral molecules or atoms within the sheath, so they lose their kinetic energy at elastic collisions. Furthermore, the VUV radiation from plasma is efficiently absorbed in atmospheric pressure—the radiation is “vacuum” ultraviolet, meaning that it propagates only under vacuum conditions.

Figure 9 represents the reported WCA of the as-treated PET samples versus the plasma treatment time. Again, the results are scattered because different authors used different discharge conditions, resulting in different fluxes of reactive plasma species on the substrates. Still, the super-hydrophilic surface finish was reported only for prolonged treatment times—100 s and above. This observation may be explained because super-hydrophilicity is obtained only on nanostructured surfaces. As explained above and according to Figure 4 and Figure 5, the nanostructuring occurs at large fluences of reactive plasma species. Significant etching is necessary for nanostructuring, and the etching increases with increasing treatment time. 

Aging causes an increase in the WCA. Figure 10 and Figure 11 represent the reported WCA after aging for 30 h. Figure 10 is the WCA versus the pressure in the plasma chamber. The trend is similar to the one in Figure 8, but some results reported the hydrophobic recovery with the WCA well above the value typical for untreated PET (between 70 and 80°). All these results are for samples with the super-hydrophilic surface finish of as-treated samples, and all are for samples aged at elevated temperatures. The paradox is explained by the loss of surface functional groups upon heating but preservation of the surface morphology.

Exciting results are revealed in Figure 11, which shows the WCA after aging for 30 h versus the plasma treatment time. The high WCAs were reported for samples aged at elevated temperatures, and the reasons for such hydrophobicity were explained above. Neglecting the results observed at high temperature, the WCA after aging for 30 h does not depend on the plasma treatment time. The results are scattered, and some measured points reveal preservation of the excellent hydrophilization. The reasons are further elaborated below.

The high wettability of plasma-treated polymers is against the law of thermodynamics, so hydrophobic recovery is a natural effect. In fact, super-hydrophilicity is not observed in nature (unlike superhydrophobicity, such as the lotus-leaf effect). The moderately low static water contact angle (between about 20 and 40°) is a consequence of the polar surface functional groups and also due to the appropriate roughness on the sub-micrometer scale. Both a high concentration of surface functional groups and the rich morphology are thermodynamically unstable. Let us first discuss the results of hydrophobic recovery reported by various authors. Many researchers have not tackled the loss of roughness of plasma-treated PET on the sub-micrometer scale, since most authors stored the plasma-treated samples at ambient conditions, i.e., room temperature. PET is rigid enough at room temperature, so the rich morphology is regarded as stable. The rigidity decreases with increasing temperature. Indeed, a polymer foil heated beyond the melting point will form a spherical droplet. The influence of temperature on the loss of rich morphology of plasma-treated PET was studied by Oh et al. [19]. They probed aging at room temperature, 40 °C, 80 °C, and 130 °C. They used PET of various crystallinities: the amorphous (A-PET) and polymer containing amorphous, ordered amorphous and crystalline phases (B-PET). The glass transition temperatures are 76 and 79 °C for A-PET and B-PET, respectively. The aging of both A-PET and B-PET at room temperature was minimal, as shown in Table 1. The almost perfect stability of the hydrophilic surface finish is explained by the peculiarities of the treatment conditions. Unlike other authors, Oh et al. [19] used energetic ions for PET treatment. According to the illustration in Figure 4, the energetic ions penetrate much deeper than O-atoms or low-energy ions. Ions cause radiation damage (i.e., displacement of atoms in the solid material due to the kinetic effect) within a few-nm-thick surface film. The implanted oxygen ions bond chemically within the layer, thus forming a few-nm-thick oxygen-rich film. The aging of samples with such a surface film cannot be through the diffusion of oxygen from the surface into the polymer, since the surface film is saturated with oxygen. The desorption of any low-molecular fragments formed on the surface during plasma treatment should occur already in the treatment chamber, since the pressure was as low as 2.6 Pa and the treatment time as long as 10 or 20 min. Furthermore, the sample bombardment with positively charged ions of the kinetic energy 400 eV (the sheath is collisionless at such a low pressure) should cause heating of the sample surface, thus enhancing desorption. Therefore, the prolonged bombardment of PET by positively charged ions assures a prolonged super-hydrophilic surface finish as long as the samples are stored at room temperature.

Figure 7 shows numerous aging curves. Curves for samples that were stored at elevated temperatures exhibit a much steeper slope. The increasing storage temperature causes faster hydrophobic recovery. Oh et al. [19] measured the concentration of various functional groups within the surface film by XPS for samples stored at elevated temperatures and reported gradual re-establishment of the original composition. They attributed the loss of oxygen upon heating to migration into the bulk, but another feasible explanation could be the desorption of low-molecular-weight fragments. Whatever the mechanism of the loss of hydrophilicity at elevated temperatures, the wettability changed from super-hydrophilic to superhydrophobic after an hour of storage at elevated temperatures. The superhydrophobic surface finish results from rich morphology and the absence of polar surface functional groups. The superhydrophobic surface finish persisted as long as the samples were heated close to the glass-transition temperature (experiments were performed at 80 °C). At a large temperature of 130 °C, however, the amorphous PET reacquired moderate hydrophobicity. SEM imaging provided by Oh et al. [19] revealed a somewhat slow but gradual flattening of the A-PET samples during storage at 130 °C. The nanostructured layer of A-PET is not rigid enough to preserve the rich morphology upon heating at temperatures between the glass transition and the melting point. 

No loss of superhydrophobic surface finish after storing plasma-treated samples at 130 °C was observed for B-PET samples (semicrystalline PET) by Oh et al. [19]. This observation indicates that the crystallinity of the surface film is at least partially preserved after treating the samples with energetic positively charged ions. The paradox may be explained by the penetration depth of ions (Figure 4), which is much smaller than the lateral dimension of the etching inhibitors (Figure 4).

As indicated in Figure 7b, other authors reported gradual hydrophobic recovery. The exemption is the report by Gotoh et al. [36], who reported almost perfectly stable WCA even after two weeks of storage at ambient conditions. No hydrophobic recovery was observed for samples treated simultaneously by plasma and VUV radiation arising from an excimer lamp [36]. Standard hydrophobic recovery was observed by Gotoh when they did not use the excimer lamp. In a way, the effect of VUV radiation is similar to the effect of positively charged ions, except that the penetration depth is larger (Figure 4): bond scission in the sub-surface PET film. As a result, the reactive oxygen species from plasma (the most important are probably O-atoms) were able to penetrate deeper into the polymer by diffusion. The consequence of simultaneous treatment with VUV radiation and plasma is the formation of a thicker modified surface layer compared to when VUV radiation is absent or negligible. The rather thick layer saturated with chemically bonded oxygen prevents diffusion of oxygen from the surface into the bulk at room temperature and, thus, prevents hydrophobic recovery. Due to stable and rather thick modified surface film, any reorientation of polar surface functional groups is also suppressed.

The upper considerations enable a qualitative explanation of the hydrophobic recovery of PET samples. As long as standard plasma treatments, such as samples at floating potential or absence of charged particles (treatment with O-atoms only) and minimal irradiation with VUV photons, are used for surface activation, the hydrophilization kinetics follow the illustration in Figure 4. The hydrophobic recovery for PET materials treated at standard conditions cannot be avoided, since the modified surface layer is very thin, and the polar groups will spontaneously orient towards the bulk during aging in order to minimize the surface energy. The hydrophobization kinetics depend on the peculiarities of the standard plasma treatments, so the hydrophobization curves (Figure 7b) differ among different treatment procedures. The hydrophobic recovery is fast during the first few hours and proceeds for several days or even weeks. The final hydrophilicity is still better than the one of untreated samples, since some polar functional groups persist on the surface for a prolonged time. The slope of all curves for aging at room temperature in Figure 7b is almost equal, which indicates similar, if not equal, kinetics of the hydrophobic recovery.

The irradiation of PET samples with energetic ions and/or VUV radiation enables more stable hydrophilicity of PET materials as long as they are stored at room temperature (or below, but not many results were reported for samples cooled upon storage). The storage temperature of 40 °C, however, facilitated hydrophobic recovery even for samples treated by energetic ions from the gaseous plasma. Increasing temperature causes rapid loss of surface functional groups and, thus, faster hydrophobic recovery.

Table 1 and Figure 7 reveal that almost all authors studied the hydrophobic recovery at the time scale of days, but the significant increase in the water contact angles occurred within the first hour or so. The insight into the hydrophobic recovery will be improved if the authors perform either experiment over a shorter period or at a lower storage temperature. The latter is often impractical, so short aging times are preferred.

Further improvement of the performance by using the proposed strategy may be achieved by carefully choosing the wavelength of the VUV radiation and the fluences. Little work has been reported on the variation of those two parameters, let alone the synergy between the radiation and the reactive oxygen species impinging on the PET surface.

## 5. Conclusions and Roadmap

The scientific literature on the hydrophobic recovery of polyethylene terephthalate was reviewed, and the observations reported by various authors were explained. The treatment of PET with oxygen-containing plasma causes surface functionalization with polar functional groups, but the surface finish depends on the peculiarities of the experimental setups. Some authors who used low-pressure oxygen plasma reported a super-hydrophilic surface finish. The hydrophobic recovery is unavoidable when using gaseous plasma with low kinetic energy of positively charged ions impinging the PET’s surface, since the modified surface film is so thin that the reorientation of the surface functional groups occurs even during storage at room temperature. The diffusion of oxygen from the surface into the subsurface layer may be a channel for hydrophobic recovery, but little work was performed on estimating the thickness of the modified surface film versus aging time. A significant loss of hydrophilicity occurs within the first hour; hence, it is recommended to measure the water contact angle for samples stored for minutes. The hydrophobic recovery increases with increasing storage temperature, but systematic experiments are yet to be performed.

The hydrophobic recovery is hardly observed at room temperature if a thicker layer of PET is modified. The formation of thicker oxygen-rich surface films can be achieved either by treatment with energetic ions or intensive sources of VUV radiation. The synergy of both treatments is yet to be studied. Oxygen plasma is a source of VUV radiation, but the penetration depth is much larger than for ions, so longer treatment times are necessary to benefit from the bond scission by VUV photons. The relatively thick modified surface film suppresses the migration or reorientation of polar functional groups, so the wettability of as-treated samples is preserved over a time scale of days, if not longer. The critical fluences of either ions, photons, or both, which enable stable hydrophilicity, are yet to be determined. The properties of the modified surface film probably also depend on the kinetic energy of positively charged ions and, thus, the DC self-biasing of samples upon treatment with capacitively coupled RF plasma and the correlations between the ion energy, surface finish, and hydrophobic recovery are yet to be evaluated. 

Finally, the surface hydrophilization and the hydrophobic recovery depends also on the crystallinity of PET samples. The samples of higher crystallinity exhibit higher stability of the surface finish, but systematic research is yet to be performed.

This review article summarizes the topic and reveals both scientific and technological challenges. The main scientific challenge is the description of the structural changes caused by vacuum ultraviolet radiation. The radiation is recommended to apply from a source while the polymer is covered with a VUV-transparent window; for example, a disc made from pure magnesium difluoride. Another scientific challenge is the determination of the role played by the photon energy. The low-pressure plasma is a suitable source of such radiation, and plasmas sustained in different gases will provide VUV radiation peaking at different wavelengths [51]. The synergies between the VUV photons and reactive oxygen species (particularly neutral oxygen atoms) also represent a scientific challenge not tackled yet in the case of PET samples. Some modeling will be useful too, but, currently, the models are limited to a few polymers and plasmas [52]. The technological challenge is the determination of the range of VUV fluences and reactive oxygen species that enable stable surface hydrophilicity. Such data are needed in any attempt to upscale the technique from laboratory experiments to industrial systems. This article only represents the guidelines, but lots of experiments will have to be performed to develop a technology for routine stable hydrophilization of polyethylene terephthalate.

## Figures and Tables

**Figure 1 polymers-14-02496-f001:**
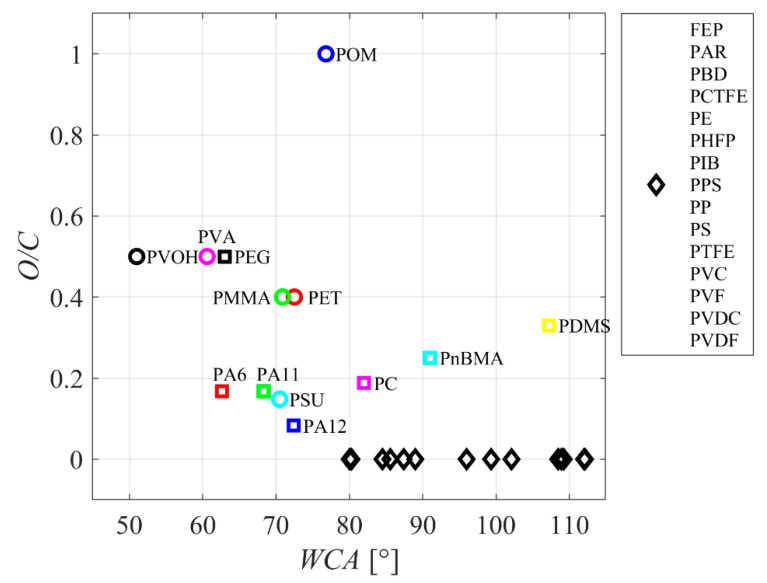
The oxygen content in polymers versus water contact angle. FEP—fluorinated ethylene propylene, PA6—Nylon 6, PA11—Nylon 11, PA12—Nylon 12, PAR—paraffin, PBD—polybutadiene, PnBMA—poly n-butyl methacrylate, PC—polycarbonate, PCTFE—polychlorotrifluoroethylene, PDMS—polydimethylsiloxane, PE—polyethylene, PEG—polyethylene glycol, PET—polyethylene terephthalate, PHFP—poly(hexafluoropropylene), PIB—polyisobutylene (butyl rubber), PMMA—polymethyl methacrylate (acrylic, plexiglass), POM—polyoxymethylene (polyacetal, polymethylene oxide), PPS—polyphenylene sulfide, PP—polypropylene, PS—polystyrene, PSU—polysulfone, PTFE—polytetrafluoroethylene, PVA—polyvinyl acetate, PVOH—polyvinyl alcohol, PVC—polyvinyl chloride, PVF—polyvinyl fluoride, PVDC—polyvinylidene chloride (Saran), PVDF—polyvinylidene fluoride. Source: [1].

**Figure 2 polymers-14-02496-f002:**
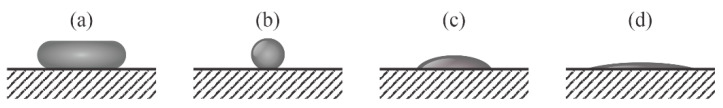
Illustration of the degree of surface hydrophilicity. A large droplet will form an oval-like shape on a superhydrophobic surface (**a**); a small water droplet will form a perfect sphere on a superhydrophobic surface (**b**); a typical shape of a small water droplet on a hydrophilic surface (**c**); a small water droplet on a superhydrophilic surface (**d**). Not to scale.

**Figure 3 polymers-14-02496-f003:**
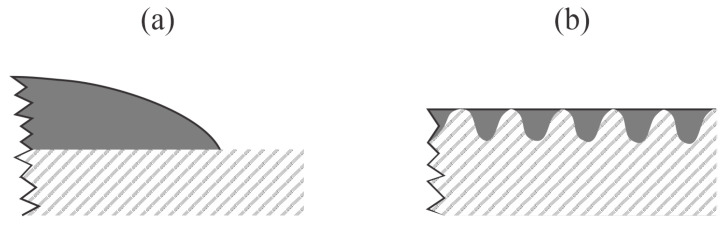
Illustration of the water droplet on a smooth hydrophilic surface (**a**) and superhydrophilic surface of rich morphology on the sub-micrometer scale (**b**).

**Figure 4 polymers-14-02496-f004:**
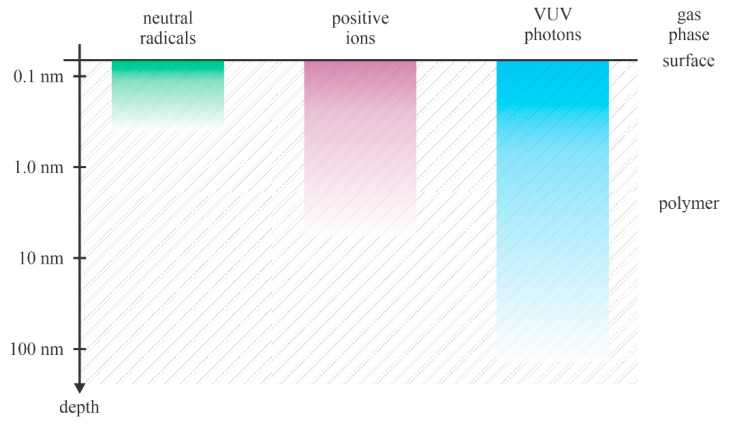
The penetration depth of neutral radicals, positively charged ions, and VUV radiation in PET.

**Figure 5 polymers-14-02496-f005:**
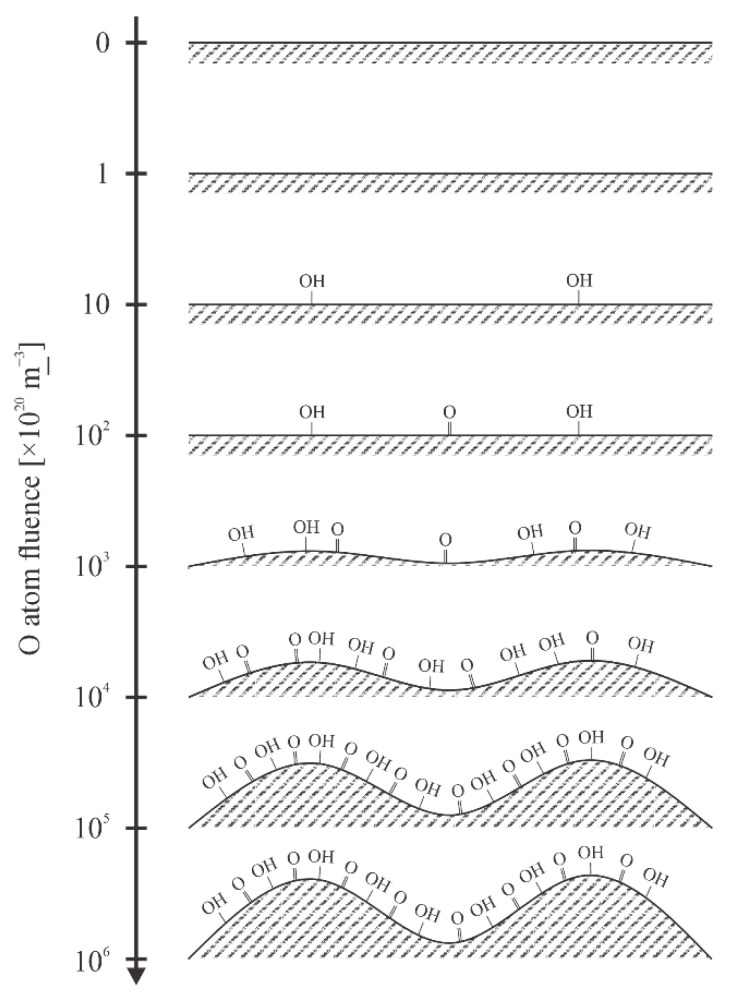
Illustration of the interaction of O-atoms with polyethylene terephthalate (not to scale). The aspect ratio between the height and the lateral dimension of the hills at the fluence of 10^25^ m^−2^ is about 0.04.

**Figure 6 polymers-14-02496-f006:**
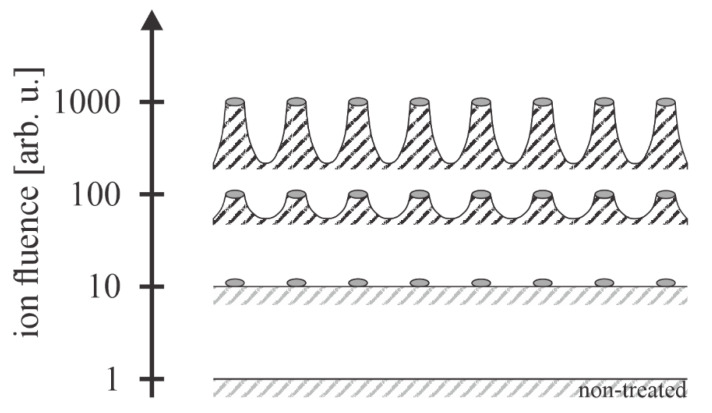
Illustration of the interaction of energetic oxygen ions with polyethylene terephthalate (not to scale). The aspect ratio between the height and the lateral dimension of the hillocks at large ion fluences may be well above 1.

**Figure 7 polymers-14-02496-f007:**
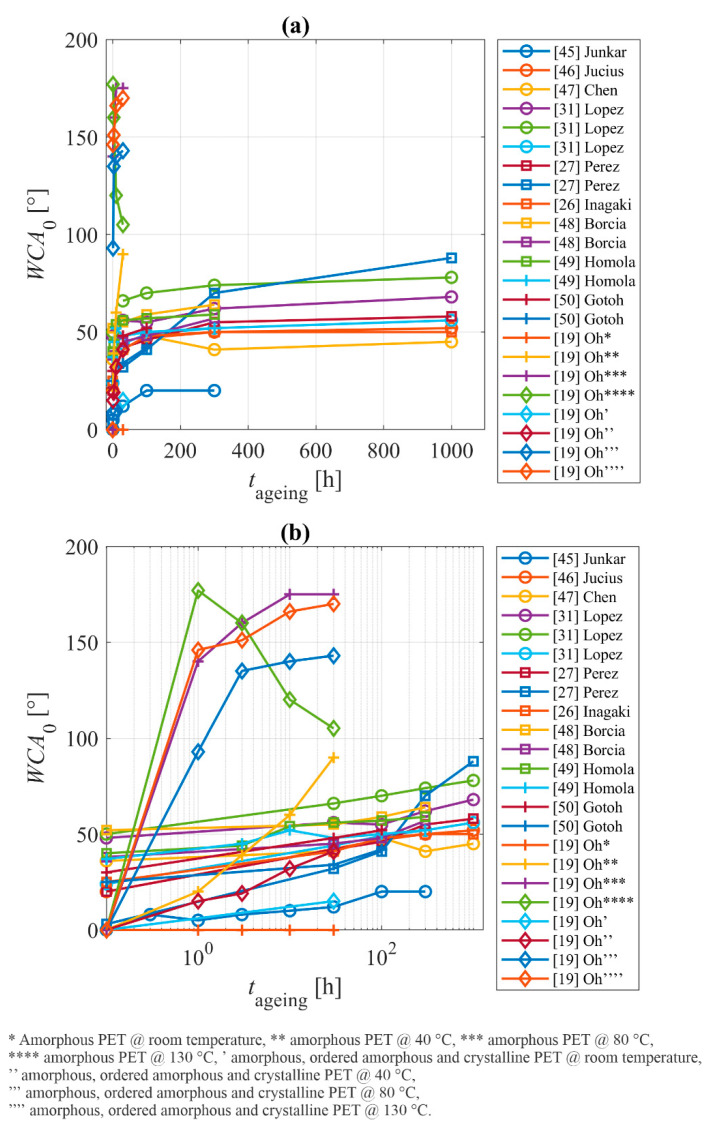
Temporal evolution of the WCA on plasma-treated PET with x-axis in linear scale (**a**) and in logarithmic scale (**b**).

**Figure 8 polymers-14-02496-f008:**
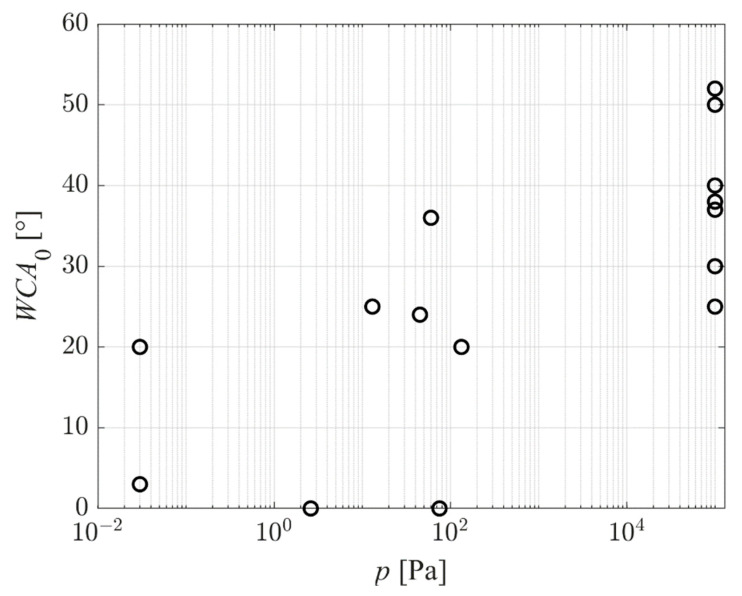
The WCA of as-treated PET samples (no aging) versus the pressure in the plasma chamber.

**Figure 9 polymers-14-02496-f009:**
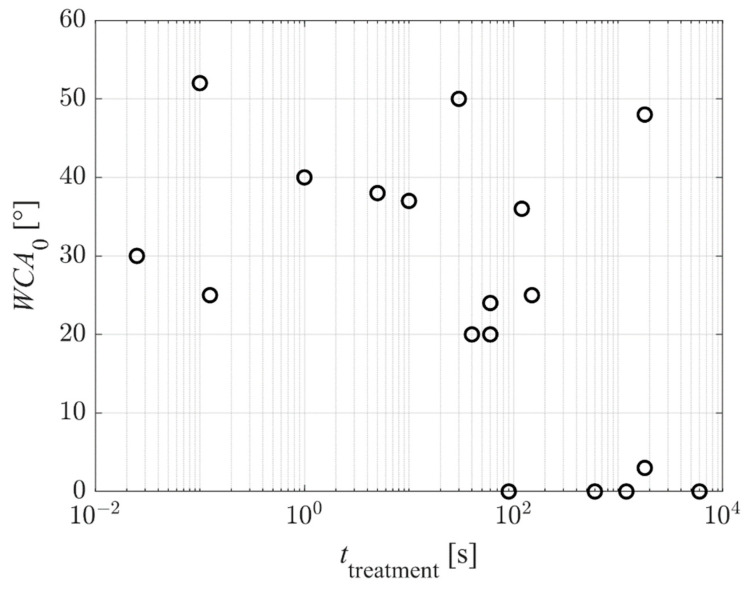
The WCA of as-treated PET samples (no aging) versus the plasma treatment time.

**Figure 10 polymers-14-02496-f010:**
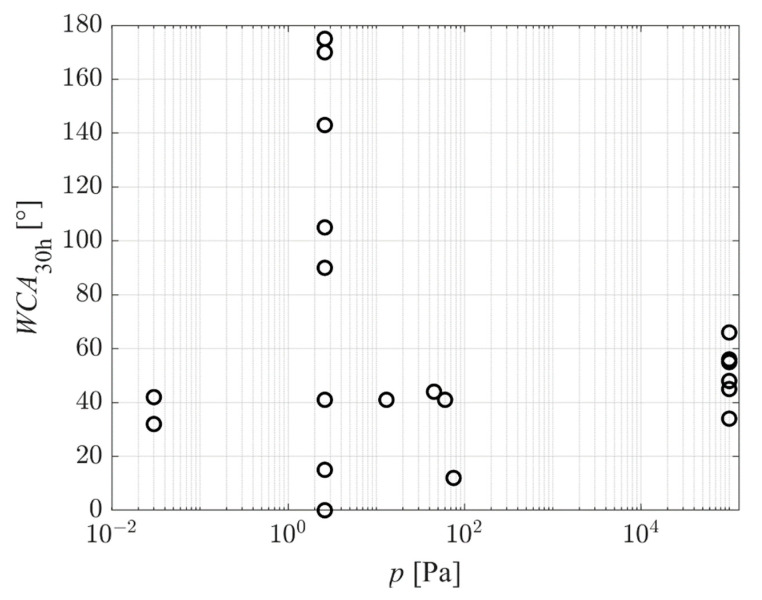
The WCA of PET samples after aging for 30 h versus the pressure in the plasma chamber.

**Figure 11 polymers-14-02496-f011:**
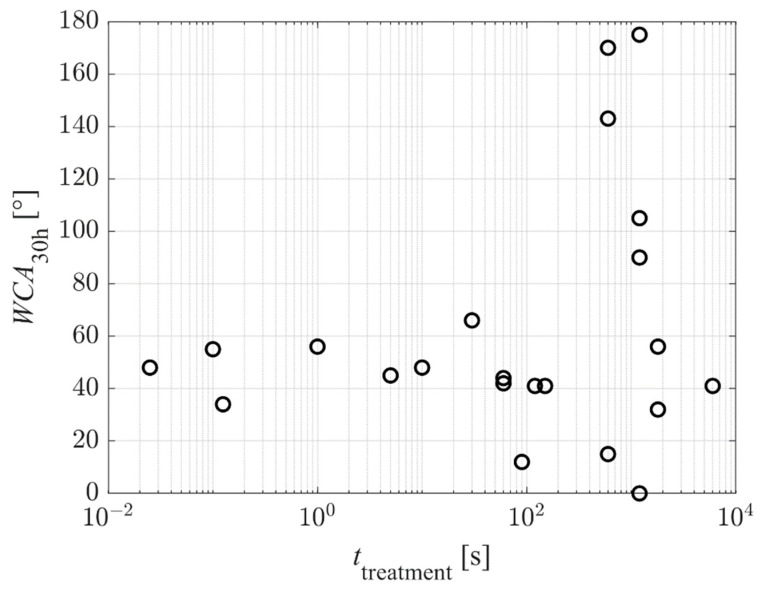
The WCA of PET samples after aging for 30 h versus the plasma treatment time.

**Table 1 polymers-14-02496-t001:** The treatment conditions and reported WCA during aging of plasma-treated PET samples.

						Static Water Contact Angle Versus Aging Time (°)								
Ref.	Author	Discharge	Power (W)	*p* (Pa)	*t*_treatment_ (s)	As-Treated	0.3 h	1 h	3 h	10 h	30 h	100 h	300 h	1000 h
[45]	Junkar	ICP RF	200	75	90	0	8	5	8	10	12	20	20	
[46]	Jucius	CCP RF	500	133	40	20						47	50	52
[47]	Chen	CCP RF	80	60	120	36					41	48	41	45
[31]	Lopez	O-atom			1800	48					56	55	62	68
[31]	Lopez	ATM DBD		Atm.	30	50					66	70	74	78
[31]	Lopez	MW	60	45	60	24					44	49	52	56
[27]	Perez	CCP RF	75	0.03	60	20					42	46	55	58
[27]	Perez	CCP RF	75	0.03	1800	3					32	41	70	88
[26]	Inagaki	CCP RF	50	13	150	25					41	48	50	50
[48]	Borcia	ATM DBD		Atm	0.1	52					55	59	64	
[48]	Borcia	ATM DBD		Atm	5	38					45	48	57	
[49]	Homola	ATM DBD		Atm	1	40			44	54	56	57	59	
[49]	Homola	ATM DBD		Atm	10	37			45	52	48	50	52	
[50]	Gotoh	ATM jet		Atm	0.025	30					48	52		
[50]	Gotoh	ATM jet		Atm	0.125	25					34	42		
[19]	Oh *	RF		2.6	1200	0		0	0	0	0			
[19]	Oh **	RF		2.6	1200	0		20	39	60	90			
[19]	Oh ***	RF		2.6	1200	0		140	160	175	175			
[19]	Oh ****	RF		2.6	1200	0		177	160	120	105			
[19]	Oh ’	RF		2.6	600	0					15			
[19]	Oh ’’	RF		2.6	600	0		15	19	32	41			
[19]	Oh ’’’	RF		2.6	600	0		93	135	140	143			
[19]	Oh ’’’’	RF		2.6	600	0		146	151	166	170			

* Amorphous PET @ room temperature, ** amorphous PET @ 40 °C, *** amorphous PET @ 80 °C, **** amorphous PET @ 130 °C, ’ amorphous, ordered amorphous and crystalline PET @ room temperature, ’’ amorphous, ordered amorphous and crystalline PET @ 40 °C, ’’’ amorphous, ordered amorphous and crystalline PET @ 80 °C, ’’’’ amorphous, ordered amorphous and crystalline PET @ 130 °C.

## Data Availability

Not applicable.

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
