# Peer review of "Hydrophobic Recovery of Plasma-Hydrophilized Polyethylene Terephthalate Polymers"

_polymers, 2022, doi:10.3390/polym14122496_

Round 1

Reviewer 1 Report

This review paper deals with interesting subject, hydrophobic recovery of polyethylene terephthalate.
However, I feel that, there are some issues that should be addressed.

1-      Add one paragraph for each mechanism in hydrophobic recovery of plasma-treated polymers

2-      Add a section about the hydrophilic and super-hydrophilic polymers

3-      Add a section about the challenges and limitation of work.

4-      Add a section about the PET

5-      The formatting of paper (subsections) and references must be as per journal guideline.

6-      Figures and Tables: Brief description on Figures and Tables required.

Figure 1: Use O/C ration in y-axis. Also the abbreviated terms in Figure 1 and others should be clarified.

Figure 5: the images is not clear. Please modified the presented image.

7-      Please add section about the durability of the plasma treated PET hydrophilic surface

8- Improve conclusion and roadmap part

Author Response

Dear Reviewer,

we found your comments useful and corrected the manuscript accordingly. Our point-by-point reply is in blue, while the modified text is in red. We believe the modified version is suitable for publication in Polymers. Please see the attachment.

Sincerely, Miran Mozetic

Reviewer 2 Report

The authors showed the mechanisms of hydrophobic recovery and methods for suppressing this effect. Also, they discussed the influence of various plasma species on the formation of the highly hydrophilic surface finish and stability of adequate wettability of PET. This manuscript is well written, but should be improved in several parts:

1.     The novelty of this manuscript is not clear enough. The authors should further highlight the novelty.

2.     The introduction is too short.

3.     If possible, the results may be supported by appropriate modelling/simulation. It will validate their discussions apart from the results from their experiment.

4.     Can the developed material system be used for LEDs as a flexible substrate? The authors should explain it in detail, otherwise this manuscript is not interesting enough (IEEE Electron Device Letters 2021, 42, 387-390; Science Bulletin 201762, 1193-1200).

5.     How to further improve the performance by using the proposed strategy? The authors are suggested to give some comments.

6.     The format of references should be carefully checked. There are some mistakes.

7.     To make this manuscript more interesting and general, related papers should be cited (e.g., Polymers 201911, 384).

Author Response

(The authors gave the same response as above.)

Round 2

Reviewer 1 Report

I have read through the revised version and response to reviewer provided by the authors, and feel the authors have done a nice job in responding to these concerns. However, the novelty of the work should be emphasis. Pleas clarify the new idea of the work at the end of introduction.

Author Response

Dear Reviewer

Thank you for your frank and positive opinion. We emphasized the novelty of this article by modifying the last paragraph of the Introduction, which now reads as:

"The new idea of the work is to examine the peculiarities of the plasma systems used by various authors to hydrophilize PET polymers. Most authors have not measured plasma species' fluxes and/or fluences and their energies but reported on the discharge parameters. A skilled plasma scientist will be able to estimate the fluxes from the reported peculiarities of the systems and thus identify the predominant species which cause the surface modifications. The penetration depth of different species differs, so it is possible to deduce the necessary treatment conditions which lead to the modification of a thicker surface film, which was found beneficial for the suppression of the hydrophobic recovery. This article focuses on the hydrophilization and hydrophobic recovery of polyethylene terephthalate and summarizes the results reported by various authors. We correlate the material wettability with the processing parameters based on available data. Such correlations have not been reported in the scientific literature, thus presenting another novelty of this article."

Sincerely